# Identification of *COL3A1*, *PLAU*, and *SPP1* as Key Biomarkers for Early Detection of Esophageal Cancer

**DOI:** 10.3390/ijms262411890

**Published:** 2025-12-10

**Authors:** Hong Zhang, Xin Cheng, Mengdi Zhang, Yixin Zuo, Shilu Zhu, Zhaorui Zuo, Xingliang Wang, Shan Lu, Xuan Gao

**Affiliations:** 1Anhui Provincial Engineering Research Centre for Molecular Detection and Diagnostics, College of Life Sciences, Anhui Normal University, Wuhu 241000, China; 18672312335@163.com; 2Anhui Provincial Key Laboratory of Molecular Enzymology and Mechanism of Major Metabolic Diseases, College of Life Sciences, Anhui Normal University, Wuhu 241000, China; cx20207778@163.com (X.C.); 15555309051@163.com (M.Z.); 19856225170@163.com (Y.Z.); 19966519371@163.com (S.Z.); zuozhaorui2024@163.com (Z.Z.); xliangw@ahnu.edu.cn (X.W.)

**Keywords:** esophageal cancer, bioinformatics, diagnostic model, random forest

## Abstract

Esophageal cancer (EC) is a highly lethal malignancy often diagnosed at advanced stages due to the lack of effective early diagnostic markers. This study aimed to identify molecular markers and construct a diagnostic model for early-stage esophageal cancer using bioinformatics approaches. Using bioinformatics, we screened three GEO datasets, locating 506 differentially expressed genes crucial to cancer progression. Our results connect ECM-receptor interaction and cytoskeleton reorganization pathways to EC. Two core gene modules came up during the protein-protein interaction analysis. From the 22 hub genes singled out, *COL3A1*, *PLAU*, and *SPP1* significantly impacted patient survival, showing considerable overexpression in cancer subjects. These genes’ expression patterns changed across cancer stages. The main novelty of our study lies in integrating these three well-known ECM-associated genes into a machine learning-based diagnostic model with an AUC of 0.98, rather than focusing on individual genes. This combined model demonstrates high diagnostic accuracy, suggesting that *COL3A1*, *PLAU*, and *SPP1* may serve as effective early-stage EC biomarkers. The diagnostic model based on these genes shows high accuracy, making it a promising tool for early-stage cancer screening.

## 1. Introduction

Esophageal cancer, primarily defined as squamous cell carcinoma and adenocarcinoma, is a highly aggressive and vicious malignancy with a five-year survival rate that peaks at 15–25%. It ranks as the sixth most deadly and ninth most common cancer worldwide [1]. Alarmingly, in China, this malignancy ranks fifth in terms of incidence and fourth in terms of mortality, making it a considerable public health issue overshadowing the nation. The low survival rate stems from the fact that most cases are diagnosed and treated at an advanced stage [2]. However, research indicates that early-stage diagnosis could significantly elevate the five-year survival rate [3], raising the urgency for the discovery and development of precise and reliable biomarkers for the early diagnosis of esophageal cancer.

Biomarker discovery for esophageal cancer (EC) has made notable progress, with several markers identified. Abnormal gene expression and mutations are key contributors to EC pathogenesis. PD-L1 expression has been used to predict response to anti-PD-1/PD-L1 therapy in EC [4]. A three-biomarker profile, including NF-kB, Gli-1, and Sonic Hedgehog, has been validated for EC prediction [5]. Overexpression of ANXA2 enhances ESCC cell migration, invasion, and metastasis through the MYC-HIF1A-VEGF pathway [6]. IPO5 promotes epithelial-mesenchymal transition (EMT) via the RAS-ERK pathway, facilitating EC progression [7]. FOXO1 is a potential prognostic factor, promoting M2 macrophage infiltration and M0 macrophage polarization [8].

However, this progress is not sufficient. These biomarkers provide some information for predicting the potential of patients with esophageal cancer to respond to immunotherapy, but there are still some limitations and challenges of practical application. PD-L1 expression can be affected by a variety of treatments, such as radiotherapy and chemotherapy, which can induce or increase PD-L1 expression. Although high TMB is associated with a good response to immunotherapy, in esophageal cancer, the proportion of patients with high TMB is low, limiting its universal use as a predictive biomarker. Although patients with high MSI generally respond better to immunotherapy, high MSI is less frequent in esophageal cancer, and few patients with esophageal cancer have been reported to have high MSI in major clinical trials. This suggests that further research is needed to improve current markers or develop new markers to more accurately guide the diagnosis and treatment of esophageal cancer. Major real-world application limitations and challenges relating to predicting the immunotherapeutic response of esophageal cancer patients still exist [9]. Improvements in current markers or the discovery of new ones is urgently needed to guide the treatment regimen more precisely. Despite previous studies linking ECM-associated genes to the progression of esophageal cancer (EC), there remains a significant gap in the establishment of a multi-gene-based diagnostic model for early detection. Furthermore, the synergistic mechanisms through which these genes interact in the early stages of tumorigenesis are still not well understood. This study aims to fill this gap by constructing a diagnostic model based on a combination of key ECM-related genes and exploring their collaborative roles in early EC development.

With the increasing richness of databases like GEO (Gene Expression Omnibus) and TCGA (The Cancer Genome Atlas), the possibility for biomarker discovery using substantial data has become more feasible [10]. In our study, we utilized the GEO database to screen and filter the different genes in samples from esophageal cancer patients and healthy individuals, an initial step towards uncovering potential biomarkers. We conducted enrichment analysis using both GO and KEGG [11], followed by network analysis with STRING [12]. Finally, visualization was carried out in Cytoscape [13], and further selection was carried out with the MCODE plugin [14]. This rigorous and systematic process allowed us to identify 382 differentially expressed genes *DEGs* and 22 hub genes. The results indicate concepts for potential therapeutic targets for esophageal cancer, with the genes identified potentially standing as substantial biomarkers for esophageal cancer that require further validation using TCGA data.

## 2. Results

### 2.1. Identification of DEGs in Esophageal Cancer

The Gene Expression Omnibus (GEO; http://www.ncbi.nlm.nih.gov/geo (accessed on 2 November 2023)) serves as a public repository that archives a wide range of high-throughput functional genomics data, including data from microarray and sequencing technologies. For our study, we focused on three gene expression datasets: GSE92396, GSE207526, and GSE194116, all accessed via GEO and utilizing the Affymetrix GPL570 platform (Affymetrix Human Genome U133 Plus 2.0 Array). The probe-to-gene symbol conversion was guided by the platform’s annotation resources. Dataset GSE92396 included 12 esophageal squamous cell carcinoma (ESCA) tissue samples, alongside 10 non-cancerous samples. Dataset GSE207526 comprised 110 ESCA samples and 10 corresponding non-cancerous samples. Dataset GSE194116 featured an additional 12 ESCA samples. Our comprehensive analysis identified a substantial number of significantly differentially expressed genes: 1566 from GSE92396, 8758 from GSE207526, and 5020 from GSE194116. Subsequent intersection analysis across these datasets revealed 506 genes that consistently exhibited differential expression, highlighting potential molecular markers for further validation and study in esophageal cancer (Figure 1).

### 2.2. Interpretation of KEGG and GO Enrichment Analysis Results for DEGs

Functional and pathway enrichment analyses were conducted using the R package clusterProfiler (v4.18.2), with gene annotations sourced from the org.Hs.eg.db database. GO term enrichment analysis revealed significant associations in biological processes, molecular functions, and cellular components. The top 20 enriched GO terms highlighted key biological processes such as epidermis development, skin development, wound healing, and extracellular matrix organization, illustrating the complex interplay of cellular activities in esophageal cancer pathogenesis (Figure 2A). Molecular functions primarily involved regulation of peptidase activity and response to wounding, indicating potential therapeutic targets. Cellular component enrichments were predominantly in the extracellular region, aligning with the observed alterations in tumor microenvironment dynamics. KEGG pathway analysis underscored the involvement of *DEGs* in critical pathways. Notably, downregulated genes were enriched in pathways like ECM-receptor interaction and the IL-17 signaling pathway, while upregulated genes showed significant presence in amoebiasis and cytoskeleton-related pathways (Figure 2B). These findings suggest a diversified yet specific alteration in pathway activities that may underpin the molecular mechanisms of esophageal cancer progression.

### 2.3. PPI Network Construction and Hub Gene Selection

Two significant core modules were identified in the PPI network (Figure 3A). The first module contained 12 hub genes: *COL1A1*, *TNFRSF11B*, *TIMP1*, *SPP1*, *SPARC*, *VCAN*, *MMP2*, *THBS2*, *COL4A1*, *SERPINH1*, *MMP9*, and *CTSK* (Figure 3B). The second module included 10 hub genes: *MMP1*, *COL3A1*, *POSTN*, *CDH17*, *MET*, *IGFBP3*, *SERPINH1*, *MMP3*, *PLAU*, and *PTGS2* (Figure 3C). These genes were found to be highly interconnected, indicating their potential key roles in the underlying biological processes related to this study. A total of 21 genes were identified as hub genes with degrees ≥ 10. The names, abbreviations and functions for these hub genes are shown in Table 1.

### 2.4. Analysis of Survival Data

Subsequent survival analysis was performed to validate the clinical significance of these hub genes. Utilizing the R package TCGAbiolinks, gene expression data and clinical information were downloaded from the TCGA database, encompassing 60,664 gene expression entries and 185 clinical records. The patients’ clinical characteristics were as follows: 108 patients were younger than 65 years, 158 were male, and the distribution of AJCC stages was as follows: 18 patients in stage I, 79 in stage II, 56 in stage III, and 9 in stage IV (Table 2). The expression levels of 22 hub genes identified from the PPI network were analyzed.

Kaplan-MeierKaplan–Meier survival analysis was conducted for all 22 hub genes identified from the PPI network. Among these, three genes—*COL3A1*, *PLAU*, and *SPP1*—were found to have a statistically significant association with overall survival (log-rank test, *p* < 0.05). In patients with high expression of these genes, survival probabilities were significantly higher compared to patients with low expression levels (Figure 4).

### 2.5. Validation and Expression Analysis of Hub Genes

The expression levels of *COL3A1*, *PLAU*, and *SPP1* were significantly higher in esophageal cancer patients compared to healthy controls (*p* < 0.05 for all three genes). As shown in Figure 5A, the boxplots illustrate the elevated expression of these genes in cancer samples compared to the control group, suggesting their potential role in tumor progression. Analysis of *COL3A1*, *PLAU*, and *SPP1* expression across different AJCC stages revealed that these genes were expressed at significantly lower levels in stage I compared to stages II, III, and IV (*p* < 0.05). This suggests that as esophageal cancer progresses to more advanced stages, these genes are upregulated, indicating their possible involvement in tumor progression (Figure 5B). For the T stage, *COL3A1*, *PLAU*, and *SPP1* expression in T1 was significantly lower than in T2, T3, and T4 (*p* < 0.05). Notably, *COL3A1* expression was significantly higher in T4 compared to all other T stages, indicating a stronger correlation between *COL3A1* expression and tumor size in advanced stages. However, *PLAU* and *SPP1* showed no significant differences between T2, T3, and T4 stages (Figure 5C). No significant differences were found in the expression of *COL3A1*, *PLAU*, and *SPP1* across the N and M stages, indicating that these genes’ expression levels do not significantly vary with lymph node involvement or distant metastasis in esophageal cancer (*p* > 0.05 for all comparisons, Figure 5D,E).

### 2.6. Development and Validation of the Diagnostic Model

The random forest model achieved an overall accuracy of 94.4% on the test set. The sensitivity of the model was 0.66667, indicating that 66.7% of esophageal cancer cases were correctly classified. The specificity was 0.96970, meaning that 96.97% of healthy controls were correctly classified as non-cancerous. These results are summarized in the confusion matrix (Figure 6A). As shown in Figure 6B, the ROC curve for the model exhibited an AUC of 0.98 with a 95% confidence interval of [0.93, 1.00]. This high AUC value indicates that the model has excellent discriminatory ability in distinguishing esophageal cancer patients from healthy controls. The feature importance analysis revealed that *COL3A1* had the highest importance value of 4.875, indicating its strong contribution to the model’s predictive performance. *PLAU* had an importance score of 3.998, followed closely by *SPP1* with a score of 3.729. These findings suggest that *COL3A1* plays a key role in the classification, followed by *PLAU* and *SPP1* (Figure 6C).

## 3. Discussion

Our study aimed to identify molecular diagnostic markers for esophageal cancer (EC) and to construct a diagnostic model with high accuracy. By analyzing genes from three large datasets, we identified pathways significantly enriched in EC, particularly ECM-receptor interaction and cytoskeleton reorganization. We discovered 22 hub genes influencing patient survival, among which *COL3A1*, *PLAU*, and *SPP1* stood out for their overexpression in cancer compared to normal tissues. These genes showed stage-dependent expression changes, suggesting their potential as biomarkers for EC. However, the key innovation of our study lies not only in identifying these three ECM-associated genes but in combining them into a diagnostic model using machine learning techniques. With an AUC of 0.98, our model demonstrates high accuracy, particularly for early-stage EC diagnosis. This integrated approach significantly enhances the diagnostic value of *COL3A1*, *PLAU*, and *SPP1*, making the combined model more clinically relevant than studying each gene in isolation.

### 3.1. Enrichment of Key Pathways in Esophageal Cancer Progression

In our study, differential gene expression analysis of esophageal cancer datasets (GSE92396, GSE207526, and GSE194116) revealed 506 common differentially expressed genes *DEGs*. GO and KEGG pathway analysis highlighted that these *DEGs* were enriched in pathways such as Amoebiasis, ECM-receptor interaction, cytoskeleton organization, and IL-17 signaling. The role of these pathways in cancer is well-established. For instance, amoebiasis can induce chronic inflammation, a condition linked to cancer development through NF-κB and IL-17 signaling. Similarly, ECM-receptor interaction is critical for tumor progression, particularly via the PI3K/AKT/mTOR and Ras/MAPK pathways, which regulate cell adhesion, migration, and proliferation. ECM receptor interaction may promote early tumor cell adhesion; cytoskeletal remodeling could enhance cell migration through the Rho/ROCK pathway, collectively facilitating the formation of a pre-metastatic niche in early-stage EC. Cytoskeleton remodeling, mediated by Ras/MAPK and PI3K/AKT pathways, plays a key role in promoting the invasive and metastatic properties of cancer cells. These findings suggest that these *DEGs* may contribute to cancer progression by facilitating cytoskeletal reorganization and enhancing cell proliferation, migration, and invasion. Our findings align with existing research on esophageal cancer (ESCC). Prior studies have shown that pathways like PI3K/AKT [15], NF-κB [16], and Ras/MAPK [17] are frequently dysregulated in ESCC, driving tumor growth, metastasis, and resistance to therapy. For instance, the PI3K/AKT/mTOR pathway has been identified as a critical axis in ESCC progression, with mutations in PIK3CA and loss of PTEN frequently observed in ESCC patients [18]. Similarly, the NF-κB pathway has been implicated in enhancing cancer cell survival, immune evasion, and metastasis. However, our study highlights the unique enrichment of genes in pathways such as amoebiasis, which is not frequently reported in ESCC literature. This suggests a potential novel link between chronic inflammation triggered by parasitic infections and esophageal cancer progression. This difference could point to novel therapeutic targets for early intervention in esophageal cancer, though further validation is required. It is clear there are certain limitations inherent to bioinformatics analysis. It begs for experimental validation to corroborate the functional roles of these newly identified *DEGs* and pathways. Hence, upcoming research endeavors ought to focus on in vitro and in vivo studies to authenticate these findings. The prime area of exploration should be investigating the intricate connection between amoebiasis-related inflammation and cancer progression.

Our study identified *COL3A1*, *PLAU*, and *SPP1* as key ECM-associated genes with potential as early diagnostic biomarkers for esophageal cancer. Beyond their well-established roles in the extracellular matrix, these genes likely play critical roles in driving early-stage tumorigenesis through distinct mechanisms. Overexpression of *COL3A1* contributes to increased ECM stiffness, which in turn enhances tumor cell migration and invasion, a hallmark of early cancer progression. The rigidity of the ECM activates mechanosensitive pathways, promoting tumor cell motility, survival, and resistance to apoptotic signals, which supports the metastatic potential of the tumor. *PLAU*, a key component of the plasminogen-plasmin system, facilitates ECM degradation by activating plasminogen to plasmin, thus breaking down the basement membrane and enabling tumor cells to invade surrounding tissues. This proteolytic activity not only allows the tumor cells to infiltrate adjacent structures but also releases bioactive molecules that support angiogenesis and inflammation, further promoting early tumor progression. Similarly, *SPP1* interacts with integrins on tumor cells and activates the PI3K/AKT signaling pathway, a crucial driver of cell survival, proliferation, and migration. This interaction enhances tumor cell adhesion and stimulates downstream signaling, promoting early-stage cell proliferation and resistance to apoptosis. Together, these mechanisms suggest that *COL3A1*, *PLAU*, and *SPP1* not only serve as promising biomarkers for early-stage esophageal cancer but also contribute to the pathophysiology of the disease by facilitating key processes such as migration, invasion, and proliferation. Our study aimed to identify ECM-related genes with diagnostic potential for esophageal cancer (EC), and we utilized datasets that included both squamous cell carcinoma (ESCC) and adenocarcinoma samples. However, it is important to note that these two histological subtypes differ significantly at the molecular level. ESCC and adenocarcinoma not only have distinct genetic alterations but also diverge in their tumor microenvironment, immune infiltration, and stromal interactions, which are critical factors influencing tumor progression and therapeutic response [19,20,21]. This distinction is particularly relevant when considering biomarkers that may perform differently across the two subtypes. Despite these differences, we believe the genes identified in our study—*COL3A1*, *PLAU*, and *SPP1*—have common roles in ECM remodeling, cell adhesion, and tumor cell migration, processes that are shared across cancer types. For instance, *COL3A1* plays a role in ECM stiffness and tumor invasion in both squamous and adenocarcinoma types [22], and *PLAU* and *SPP1* [23] have been implicated in promoting tumor cell migration and metastasis via interactions with integrins and the PI3K/AKT pathway [24]. However, given the molecular and microenvironmental differences between ESCC and adenocarcinoma, we acknowledge that further validation is needed. To strengthen the robustness and generalizability of our diagnostic model, future studies should focus on stratifying these two subtypes separately and validating the proposed biomarkers in homogeneous cohorts.

### 3.2. Identification of Hub Genes and Their Prognostic Significance

We identified two key modules through PPI analysis of esophageal cancer datasets. Survival analysis revealed three significant genes: *COL3A1*, *PLAU*, and *SPP1*. These genes influence crucial pathways in cancer progression, particularly in esophageal cancer. *COL3A1* has been reported to be overexpressed in various cancers, including breast [25] and colorectal cancers [26], where it contributes to ECM remodeling and enhances tumor invasiveness. Similarly, *PLAU* plays a critical role in ECM degradation and has been linked to poor prognosis in several cancers, including head and neck squamous cell carcinoma (HNSCC) [27] and lung cancer [28], through its role in promoting metastasis. *SPP1*, a known mediator of tumor microenvironment interactions, has been shown to be highly expressed in multiple cancers, including breast [29], lung [30], and colorectal [31] cancers, where it enhances tumor growth, invasion, and immune evasion. However, our study uniquely highlights the significance of these genes specifically in esophageal cancer, an area where research on their roles has been comparatively sparse. The observation that these three genes correlate significantly with patient survival in esophageal cancer adds a new dimension to their biological importance. However, *COL3A1*, *PLAU*, and *SPP1* show higher expression in tumors compared to normal tissues, yet lower expression of these genes in tumors is associated with worse clinical outcomes, warrants further consideration. One possible explanation lies in the complex roles these genes may play at different stages of cancer progression. While increased expression of ECM-related genes like *COL3A1*, *PLAU*, and *SPP1* may initially contribute to tumor growth, invasion, and metastasis by promoting ECM remodeling and cell migration, their downregulation later in tumor progression might reflect a more aggressive, metastatic phenotype. For instance, lower expression of these genes could indicate a shift toward a more invasive state, where the tumor cells have adapted to escape the primary site and invade surrounding tissues. While previous studies have identified these genes in other cancers, their role in esophageal cancer, particularly their involvement in survival outcomes, may offer new therapeutic targets and diagnostic markers. Future research could look at gene interactions with other pathways such as PI3K/AKT or NF-κB, giving a broader understanding of their involvement in cancer biology. This underscores the necessity to investigate ECM-related genes in cancer research.

### 3.3. Development of a Diagnostic Model for Early-Stage Esophageal Cancer Based on COL3A1, PLAU, and SPP1

We built a random forest algorithm-based diagnostic model, achieving a high 0.98 AUC, showing its accuracy and reliability. This is crucial as early detection improves survival rates. These genes may offer potential for developing non-invasive diagnostics.

Several studies have explored molecular markers and machine learning models for the diagnosis of esophageal cancer. For instance, models using DNA methylation markers like SEPT9 have shown good specificity but have relatively low sensitivity for early-stage EC detection [32], with sensitivities ranging between 58.8% to 79.2% depending on the studythis study. Additionally, advanced machine learning models based on imaging modalities such as gastroscopy and narrow-band imaging have achieved high accuracies for EC detection, with some models reporting accuracies as high as 98%. Compared to these approaches, our model’s performance stands out due to its high AUC of 0.98, suggesting better diagnostic potential, especially for early-stage detection. While the existing models using DNA methylation and imaging have made significant strides, they often struggle with early-stage sensitivity. Our gene-based model addresses this gap, providing more reliable diagnostics at the crucial early stage of the disease. We identified *COL3A1*, *PLAU*, and *SPP1* as early-stage esophageal cancer markers, highlighting the role of these genes in disease progression. Their incorporation into a high-accuracy diagnostic model could transform early cancer detection and boost patient outcomes. However, this model, largely dependent on computational analysis, requires further validation in real-world clinical environments. Future research should focus on clinical trial validations and explore integrating this model with other diagnostic methods. This gene-based model shows potential as a game-changer for esophageal cancer screening strategy enhancement.

Based on this study, it is crucial we expand research in several areas. Firstly, validating the identified biomarkers for early-stage esophageal cancer in larger clinical groups is necessary. Further study, including in vitro and in vivo experiments, will confirm roles of these genes in cancer progression. Combined use of biomarkers with methods such as DNA methylation panels and non-invasive liquid biopsies may improve diagnostics. It is also essential to deepen understanding of the molecular mechanisms of esophageal cancer progression. Lastly, a multi-omics approach will enhance our understanding of esophageal cancer biology, enabling personalized treatment. A notable paradox in our findings is that the three selected genes, *COL3A1*, *PLAU*, and *SPP1*, exhibit higher expression in tumors compared to normal tissues, yet lower expression of these genes within tumors is associated with worse clinical outcomes. This finding warrants further discussion, particularly in the context of tumor heterogeneity, which is not captured in bulk gene expression analyses. Bulk RNA sequencing represents the average gene expression across all cell types in a tumor sample, but tumors are complex and contain a variety of cell populations, each with distinct molecular signatures. As such, gene expression in bulk tumor samples may mask differences between tumor cell subpopulations, immune cells, and stromal cells, each of which can influence tumor progression differently. One possible explanation for this paradox is the intratumoral heterogeneity of gene expression. While *COL3A1*, *PLAU*, and *SPP1* may be highly expressed in certain tumor populations (e.g., stromal or fibroblastic cells), the downregulation of these genes in other subpopulations of the tumor, particularly those that are more invasive or metastatic, could reflect a shift towards a more aggressive phenotype. For instance, *SPP1* high expression in LUAD is associated with poor prognosis, and its downregulation can inhibit cell migration and invasion while altering the expression of EMT markers [33]. *COL3A1* can regulate the immunosuppressive microenvironment in glioma and participate in the EMT process. Knockdown of *COL3A1* significantly inhibits the migration and invasion ability of glioma cells [34]. This dynamic role of ECM-related genes, potentially acting at different stages of tumor progression, might explain the inverse relationship between expression levels and clinical outcomes [35]. To explore this hypothesis more effectively, future studies should consider the use of single-cell RNA sequencing (scRNA-seq) or spatial transcriptomics, which are increasingly employed to study tumor microenvironment heterogeneity at a high resolution. These methods allow for a more granular understanding of gene expression within individual cell types in the tumor and could clarify whether *COL3A1*, *PLAU*, and *SPP1* are differentially expressed across various cell types within the same tumor. The heterogeneity of gene expression within tumors is a critical factor to consider in understanding how these biomarkers influence tumor progression and patient prognosis. We acknowledge that this complexity is a key consideration when interpreting bulk gene expression data, and future studies incorporating single-cell analyses will be crucial to resolve the paradox observed in our study.

### 3.4. Study Limitations and Significance

Despite the promising findings, this study has several limitations. The public datasets we utilized (GSE92396, GSE194116, and GSE207526) lacked detailed pathological stage information for the individual patient samples. This makes it challenging to conclusively determine whether these markers are truly applicable for early detection, as early-stage tumors were not separately identified in the datasets. Nevertheless, our analysis of the TCGA data has demonstrated that *COL3A1* and *SPP1* show significant differential expression across AJCC tumor staging, and *COL3A1* and *PLAU* show significant differences in T-staging, while *COL3A1* and *SPP1* are differentially expressed in N-staging. These findings suggest that these biomarkers may have the potential to distinguish between early and late-stage esophageal cancer. However, these results are preliminary, and further validation in well-characterized clinical cohorts, specifically with early-stage tumor samples, would be necessary to confirm their suitability for early detection. One limitation of our study is the inability to obtain patient samples for experimental validation. This represents a major constraint, and we recognize that true early detection typically involves non-invasive methods, such as liquid biopsies or imaging, which are not directly addressed in our study. As our findings are based on bioinformatic analyses of existing datasets, we urge caution in the interpretation of our results. We emphasize that while *COL3A1*, *SPP1*, and *PLAU* show promise as biomarkers for distinguishing between different stages of esophageal cancer, further prospective studies with stage-specific validation, ideally using clinical samples, are required to substantiate their role in early detection and patient management.

## 4. Materials and Methods

### 4.1. Sample Collection and Processing

For this investigation, we utilized microarray data retrieved from the Gene Expression Omnibus (GEO), a highly regarded public repository that archives a vast collection of functional genomics data. The specific platform engaged for this analysis was the Affymetrix GPL570 (Affymetrix Human Genome U133 Plus 2.0 Array). It facilitated us in procuring three distinct gene expression datasets: GSE92396, GSE207526, and GSE194116. It is worth noting that all the probes gleaned from the platform were scrupulously translated into their equivalent gene symbols, in sync with the relevant annotation details explicitly provided. The GSE92396 dataset was comprised of a relatively small collection, 12 tissue samples obtained from patients afflicted with esophageal carcinoma (ESCA), juxtaposed with 10 samples sourced from non-cancerous tissues. Conversely, the GSE207526 dataset was more expansive, housing 110 ESCA tissue samples, and again, 10 samples from non-cancerous origins. The GSE194116 dataset was dedicated exclusively to ESCA, containing 12 tissue samples. In sum, the careful stratification and thorough examination of these datasets power our persistent quest for a deeper understanding of esophageal carcinoma.

To address batch effects, we employed the ComBat method from the sva R package, which is specifically designed for batch effect correction in gene expression data. This step helps ensure that any systematic variations between the datasets due to different experimental conditions do not bias our results. Normalization of gene expression data across different datasets was performed using the quantile normalization method to make the datasets comparable.

### 4.2. Identification of DEGs

Differential Expression Genes *DEGs* Identification. The *DEGs* distinguishing between esophageal squamous cell carcinoma (ESCA) and non-cancerous specimens were meticulously analyzed utilizing GEO2R (http://www.ncbi.nlm.nih.gov/geo/geo2r (accessed on 5 November 2023)), an interactive web-based tool. GEO2R convincingly enables the comparison of two or more datasets in a GEO series to detect *DEGs* across contrasting experimental conditions. Each dataset underwent quantile normalization to account for technical variations and ensure comparability across datasets. Following normalization, log2 transformation was applied to the expression data. Differentially expressed genes (DEGs) were identified using the criteria of |log2-fold change| > 1 and an adjusted *p*-value < 0.05. To account for multiple testing, False Discovery Rate (FDR) correction was applied to adjust the *p*-values. Subsequently, the analytical examination of the intersecting results was conducted using the R package.

### 4.3. KEGG and GO Enrichment Analyses of DEGs

We conducted a functional enrichment analysis on differentially expressed genes *DEGs*, utilizing a series of rigorously compiled and biologically relevant databases. We employed DAVID: Functional Annotation Tools, an acclaimed online biological information repository (https://david.ncifcrf.gov/tools.jsp (accessed on 8 November 2023)) recognized for its integration of biological information and sophisticated software for analytical interpretation. Furthermore, we incorporated data from KEGG, a prime database resource renowned for facilitating comprehension of elevated functions and intricate biological systems drawn from voluminous molecular datasets elicited through advanced, high-throughput experimentation techniques. As a testament to our comprehensive analytical approach, we also utilized the Gene Ontology (GO) initiative, an internationally utilized bioinformatics tool that annotations of genes, provides insightful scrutiny into the biological implications of these genes. GO meticulously classifies gene function terms into three distinct ontologies: Biological Process (BP), Cellular Component (CC), and Molecular Function (MF). For a thorough dissection of *DEG* functionalities, we employed the DAVID online database for executing detailed biological analyses. Observations where *p* < 0.05 were deemed statistically substantial, hence meriting further consideration. The *DEGs* underwent extensive exploration via precise GO and purposeful KEGG analyses.

### 4.4. PPI Network Construction and Module Analysis

Our research delved meticulously into the structure and module analysis of the Protein-ProteinProtein–protein Interaction (PPI) network. This analysis was strategically carried out, relying extensively on an interactome obtained from the STRING-db.org online database (version 10.0). The interactome was chosen for its reliable reputation in predicting network interacting genes. It is of paramount importance to scrutinize the functional interactions between proteins to cogently derive a well-functioning disease mechanism. In light of this, this research uncovers the construction of the PPI network of differentially expressed genes *DEGs*. The STRING database was exploited for this purpose, specifically focusing on interactions that yielded a composite score surpassing 0.4; these were deemed statistically significant. Visual exploration of the molecular interaction networks was accommodated using the open-source bioinformatics software platform known as Cytoscape (version 3.4.0)—an authoritative tool for our data visualization. The meticulous process of our research design and execution ensures a competent overview of the subject matter, emphasizing its applicability and reliability. For module analysis, the Molecular Complex Detection (MCODE) plugin in Cytoscape was used to identify densely connected regions within the PPI network. The degree cutoff for hub gene identification was set to ≥10 to ensure the selection of highly interactive genes.

### 4.5. Hub Genes Selection and Analysis

The final selection of hub genes, which amounted to 7 in this case, was performed using the CBIO portal (https://www.cbioportal.org/ (accessed on 11 November 2023)). The biological process analysis and visualization were carried out using the Cytoscape biological network gene oncology tool (Bingo, version 3.0.3). Furthermore, the overall survival of the hub genes was evaluated using Kaplan-MeierKaplan–Meier curves derived from the cBioPortal. This methodology presents an authoritative, thorough and rigorous approach to disease gene profiling, paving the way towards the development of more effective diagnostic and therapeutic strategies for various cancer types.

To corroborate the clinical significance of the hub genes identified, we undertook an imperative examination of the differential genes with the associated survival data. By employing Kaplan-MeierKaplan–Meier methodology and the Cox proportional hazard model, we were able to systematically appraise the correlation between gene expression levels and patient survival duration. Survival analysis as an approach to identifying hub genes represents a key methodology and standard within this study. Leveraging the comprehensive transcriptome data available in the UALCAN database for The Cancer Genome Atlas (TCGA), this approach facilitates the detection of biomarkers and the evaluation of gene expression across different molecular subtypes. UALCAN is a powerful tool, offering graphical representation of gene expression, patient survival information based on gene expression, and the ability to identify novel diagnostic and therapeutic targets. By analyzing survival in differentially expressed genes *DEGs* using UALCAN, the overall survival rate can be determined. Genes displaying a *p*-value under 0.05 are considered statistically significant and are therefore primarily screened as hub genes. In addition, the UALCAN database offers a unique insight into disease progression, with the potential to analyze gene expression shifts at different disease stages. This aspect, represented by the clinical data sample of esophageal cancer (ESCA) within the UALCAN database, underlines the potential for more accurate disease staging. Using UALCAN, the roles of identified hub genes were explored in the context of disease progression in ESCA.

### 4.6. Survival Analysis

Survival analysis was conducted using clinical data from the TCGA-ESCA cohort, which was downloaded using the TCGAbiolinks R package. A comprehensive statistical analysis of clinical data was conducted, characterizing the samples by age, gender, AJCC stage, race, and TNM stages. Overall survival (OS) was used as the survival outcome, and Kaplan-MeierKaplan–Meier survival curves were employed to compare survival probabilities between high and low expression groups for each gene. Patients were stratified into high and low expression groups based on the median expression level of each gene. The log-rank test was used to evaluate the statistical significance of survival differences between these groups. Cox proportional hazards regression analysis was also performed to assess the impact of each hub gene on patient survival. All analyses were performed using the survival package in R (version 4.3.1). The Kaplan-MeierKaplan–Meier survival curves and Cox regression models were generated for all hub genes, and genes with a log-rank *p*-value < 0.05 were considered statistically significant.

### 4.7. Validation of Hub Genes and Differential Expression Analysis

Gene expression data were obtained from the publicly available GSE207526 dataset in the GEO database. This dataset includes gene expression profiles from 110 esophageal cancer patients and 10 healthy controls. In addition, gene expression data related to different clinical stages of esophageal cancer were downloaded from the TCGA-ESCA dataset using the TCGAbiolinks R package. The clinical stages analyzed included the AJCC stage, T stage (tumor size), N stage (lymph node involvement), and M stage (metastasis). The expression levels of hub genes were extracted from the GSE207526 dataset using the GEOquery R package. These hub genes were selected for validation due to their significant role in esophageal cancer based on previous PPI network analysis.

To compare the expression differences of hub genes between esophageal cancer patients and healthy controls, a t-test was performed. A *p*-value < 0.05 was considered statistically significant. To evaluate the differences in gene expression levels across different stages of esophageal cancer, including the AJCC stages (I, II, III, IV), T stages (T1, T2, T3, T4), N stages (N0, N1, N2, N3), and M stages (M0, M1), ANOVA (analysis of variance) was applied to compare gene expression across these stages. A significance level of *p* < 0.05 was set for these analyses. ggplot2 was used to generate boxplots for visualizing the gene expression differences between cancer patients and healthy controls, as well as across different cancer stages (AJCC, T, N, M stages).

### 4.8. Diagnostic Model Construction and Validation

Gene expression data were obtained from the publicly available GSE207526 dataset, which includes expression profiles from 110 esophageal cancer patients and 10 healthy controls. The dataset was randomly split into two parts: 70% for training (77 cancer patients and 7 healthy controls) and 30% for testing (33 cancer patients and 3 healthy controls). A random forest classifier was constructed using the expression levels of the *COL3A1*, *PLAU*, and *SPP1* genes to predict whether a sample belonged to an esophageal cancer patient or a healthy control. The model was implemented in R using the randomForest package. The number of decision trees was set to 500, and the model was trained on the training dataset. We applied k-fold cross-validation with 10 folds during model training to ensure the generalizability and robustness of the model. The model’s performance was evaluated on the test dataset. A confusion matrix was used to calculate the accuracy, sensitivity, and specificity of the model. Additionally, a receiver operating characteristic (ROC) curve was plotted, and the area under the curve (AUC) was computed using the pROC package in R to assess the overall performance of the model. Feature importance was assessed using the importance function from the randomForest package. The importance of each gene was calculated based on the mean decrease in Gini index, which reflects the contribution of each gene to the model’s predictive power. We also accounted for multiple testing and adjusted *p*-values using the False Discovery Rate (FDR) method as proposed by Benjamini-Hochberg.

## 5. Conclusions

This study screened three GEO datasets, locating 506 differentially expressed genes crucial to cancer progression. The results connect ECM-receptor interaction and cytoskeleton reorganization pathways to EC. Two core gene modules came up during the protein-proteinprotein–protein interaction analysis. From the 22 hub genes singled out, *COL3A1*, *PLAU*, and *SPP1* significantly impacted patient survival, showing considerable overexpression in cancer subjects. These genes’ expression patterns changed across cancer stages. Using these genes, we constructed a machine learning-based model with high reliability exhibiting an AUC of 0.98, demonstrating its potency in diagnosing early-stage EC. The diagnostic model based on these genes shows high accuracy, making it a promising tool for early-stage cancer screening.

## Figures and Tables

**Figure 1 ijms-26-11890-f001:**
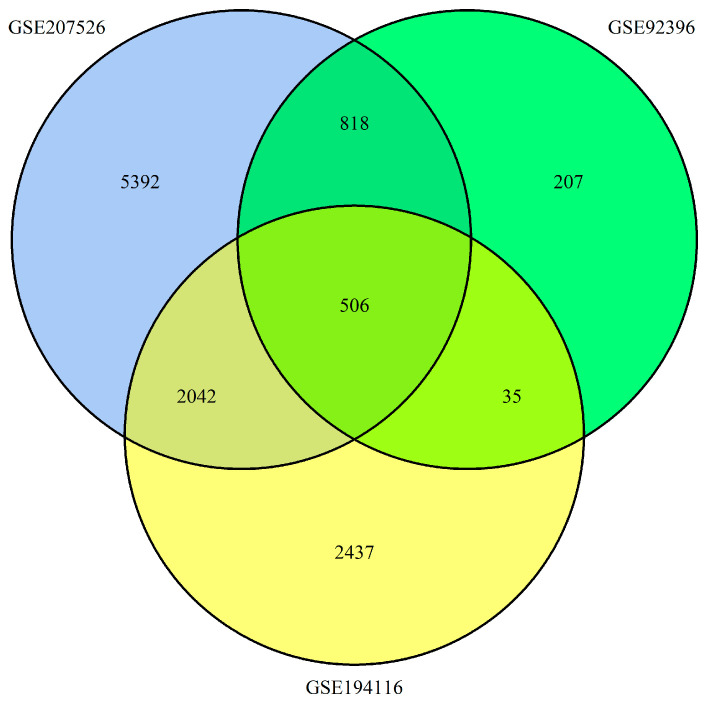
Venn diagram of *DEGs* which were selected with a fold change > 1 and *p*-value < 0.01 among the mRNA expression profiling sets GSE92396, GSE207526, and GSE194116. Of the identified DEGs, 241 genes were upregulated and 265 genes were downregulated.

**Figure 2 ijms-26-11890-f002:**
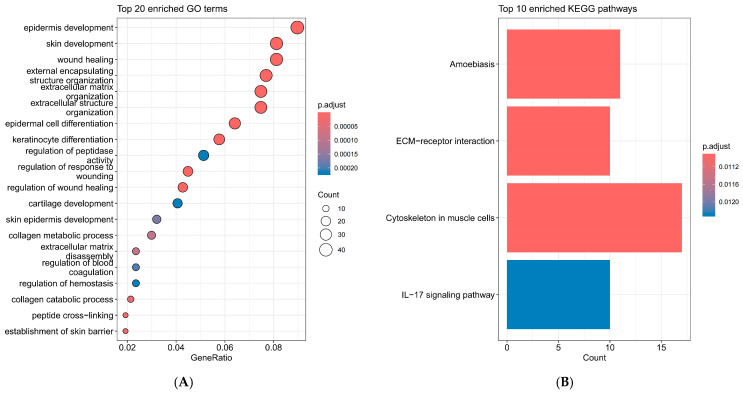
GO and KEGG pathway enrichment analysis of *DEGs* in the most significant module. (**A**) GO analysis. (**B**) KEGG pathway analysis.

**Figure 3 ijms-26-11890-f003:**
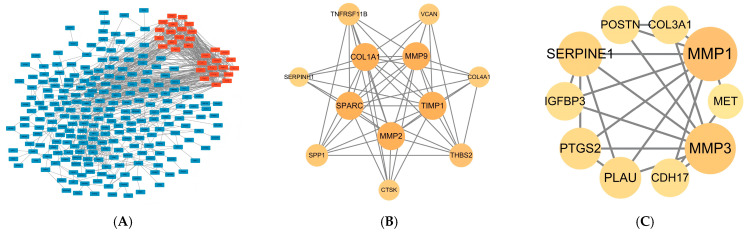
Visualization of the PPI Network in Cytoscape (**A**) protein-proteinprotein–protein interaction (PPI) network constructed from differentially expressed genes in esophageal cancer. (**B**) the first significant module extracted using the MCODE plugin in Cytoscape. (**C**) the second significant module extracted, featuring the second set of hub genes. The network diagram shows a subset of genes, with the genes of primary focus highlighted in deep red.

**Figure 4 ijms-26-11890-f004:**
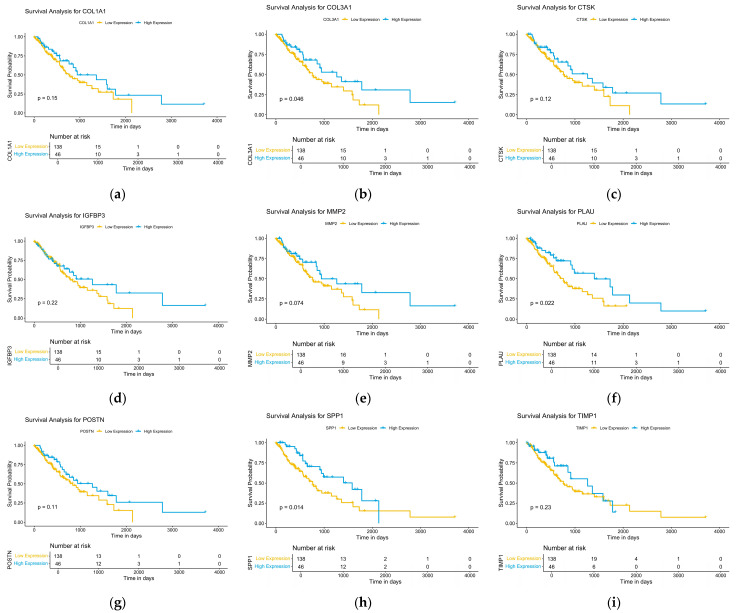
Overall survival analyses of hub genes were performed using R packages. *p* < 0.05 was considered statistically significant. (**a**) COL1A1; (**b**) COL3A1; (**c**) CTSK; (**d**) IGFBP3; (**e**) MMP2; (**f**) PLAU; (**g**) POSTN; (**h**) SPP1; (**i**) TIMP1.

**Figure 5 ijms-26-11890-f005:**
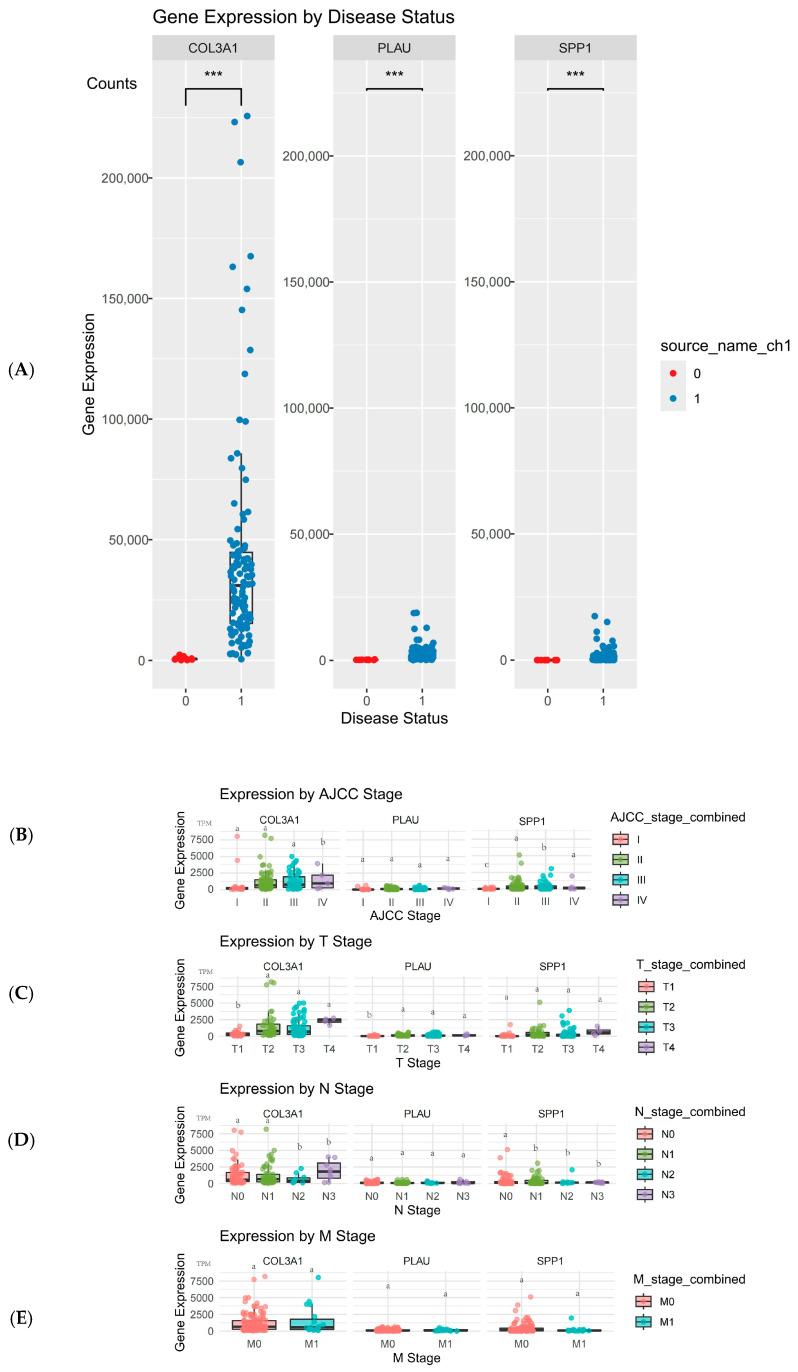
Expression Level Analysis of *COL3A1*, *PLAU*, and *SPP1* in cancer tissues and normal tissues. (**A**) Boxplot analyses were performed to assess the expression levels of the genes *COL3A1*, *PLAU*, and *SPP1* in cancer patients’ tissues (represented by 1) and normal tissues (represented by 0); (**B**) Comparison of expression levels of the three genes across different AJCC stages; (**C**) Comparison of expression levels of the three genes across T stages; (**D**) Comparison of expression levels of the three genes across N stages; (**E**) Comparison of expression levels of the three genes across M stages. Statistical significance is indicated by ‘***’ and letters (a, b, c), where ‘***’ represents a significance level of *p* < 0.001, and the letters indicate significance at *p* < 0.05.

**Figure 6 ijms-26-11890-f006:**
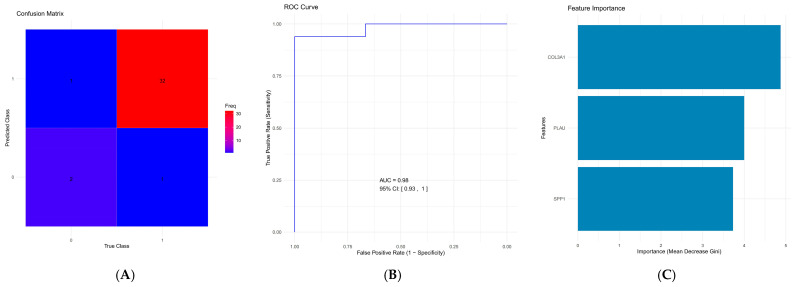
The random forest model’s performance in classifying esophageal cancer. (**A**). The model’s ability to discriminate between esophageal cancer patients and healthy controls is showcased by the ROC curve with an AUC of 0.98 (**B**). Feature importance scores are highlighted, with *COL3A1* being the most influential (4.875), followed by *PLAU* (3.998) and *SPP1* (3.729), indicating their significant contribution to the model’s predictive performance (**C**).

**Table 1 ijms-26-11890-t001:** The Symbol, abbreviations and functions for 21 hub genes.

Gene Symbol	Full Name	Function
*CDH17*	Cadherin 17	Mediates calcium-dependent cell-cellcell–cell adhesion in the intestines
*COL1A1*	Collagen, Type I, Alpha 1	Produces type I collagen, which is critical for skin, bone, tendon, and other connective tissues
*COL3A1*	Collagen, Type III, Alpha 1	Provides structure to tissues such as skin, lung, and vascular tissues
*COL4A1*	Collagen, Type IV, Alpha 1	A major component of basement membranes, essential for tissue structure and function
*CTSK*	Cathepsin K	A protease involved in the breakdown of collagen in bones, critical for bone resorption
*IGFBP3*	Insulin-Like Growth Factor Binding Protein 3	Regulates the availability of insulin-like growth factors, involved in cell growth and apoptosis
*MET*	MET Proto-Oncogene, Receptor Tyrosine Kinase	Encodes a protein involved in cell growth, development, and wound healing
*MMP1*	Matrix Metallopeptidase 1	Breaks down interstitial collagens, involved in tissue remodeling and repair
*MMP2*	Matrix Metallopeptidase 2	Degrades type IV collagen in the ECM, involved in tissue remodeling and metastasis
*MMP3*	Matrix Metallopeptidase 3	Degrades various components of the ECM, involved in tissue remodeling
*MMP9*	Matrix Metallopeptidase 9	Degrades components of the ECM, involved in tissue remodeling, inflammation, and cancer metastasis
*PLAU*	Plasminogen Activator, Urokinase	Converts plasminogen to plasmin, involved in breaking down blood clots and tissue remodeling
*POSTN*	Periostin	Involved in tissue repair, bone development, and maintaining the integrity of the ECM
*PTGS2*	Prostaglandin-Endoperoxide Synthase 2	An enzyme involved in inflammation and pain, also known as COX-2
*SERPINH1*	Serpin Family H Member 1	A collagen chaperone involved in collagen biosynthesis and folding
*SPARC*	Secreted Protein, Acidic, Cysteine-Rich (Osteonectin)	Regulates cell-matrix interactions, cell migration, and tissue remodeling
*SPP1*	Secreted Phosphoprotein 1 (Osteopontin)	Involved in bone remodeling, immune responses, and cell adhesion
*THBS2*	Thrombospondin 2	Regulates angiogenesis, tissue repair, and cell adhesion
*TIMP1*	Tissue Inhibitor of Metalloproteinases 1	Inhibits matrix metalloproteinases (MMPs) and regulates ECM degradation, cell growth, and apoptosis
*TNFRSF11B*	Tumor Necrosis Factor Receptor Superfamily Member 11b	Acts as a decoy receptor for RANKL, inhibiting osteoclastogenesis and bone resorption
*VCAN*	Versican	A large extracellular matrix proteoglycan involved in cell adhesion, proliferation, and ECM organization

**Table 2 ijms-26-11890-t002:** The patients’ clinical characteristics.

Clinical Characteristics		N	(%)
Age(years)	<65	108	58.4
	≥65	77	41.6
Gender	Male	158	85.4
	Female	27	14.6
Race	Asian	46	24.9
	American	5	2.7
	White	114	61.6
	Not report	20	10.8
AJCC Stage	I	18	9.7
	II	79	42.7
	III	56	30.3
	IV	9	4.9
T classification	T0	1	0.5
	T1	31	16.8
	T2	43	23.2
	T3	88	47.6
	T4	5	2.7

## Data Availability

The datasets analyzed during the current study can be found in the GEO database, https://www.ncbi.nlm.nih.gov/geo/query/acc.cgi?acc=GSE92396 (accessed on 5 November 2023), https://www.ncbi.nlm.nih.gov/geo/query/acc.cgi?acc=GSE207526 (accessed on 5 November 2023), https://www.ncbi.nlm.nih.gov/geo/query/acc.cgi?acc=GSE194116 (accessed on 5 November 2023).

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
