# Peer review of "Identification of *COL3A1*, *PLAU*, and *SPP1* as Key Biomarkers for Early Detection of Esophageal Cancer"

_ijms, 2025, doi:10.3390/ijms262411890_

Round 1
Reviewer 1 Report
Comments and Suggestions for Authors
The manuscript explores potential biomarkers for early esophageal cancer detection using integrative bioinformatics analysis. From three GEO datasets and TCGA validation, the authors identified 382 DEGs and 22 hub genes, highlighting COL3A1, PLAU, and SPP1 as survival-associated markers. The study addresses an important clinical challenge early detection of esophageal cancer through data mining. The analytical pipeline is well-structured and reproducible. However, the absence of experimental or external dataset validation, limited biological contextualization, and potential overfitting of the model restrict its translational impact. Substantial revisions are required before consideration for publication. Strehgth of this manuscripts are:
- Comprehensive dataset integration (three GEO datasets + TCGA validation) enhances robustness.
- Systematic multi-step analysis (DEG, GO/KEGG, PPI, survival, machine learning) demonstrates technical competence.
- Clear visual presentation (Venn diagrams, survival plots, ROC curves) aids interpretability.
- The diagnostic model shows excellent in silico performance (AUC = 0.98).
Following are some of the weaknesses associated with this manuscript.
- The findings remain computational; without qPCR, IHC, or external dataset validation, diagnostic relevance is speculative.
- COL3A1, PLAU, and SPP1 are well known ECM associated genes. The main novelty should be clarified as the combined diagnostic modeling for EC.
- Statistical errors for exampls; lack of batch correction or normalization across datasets; Missing adjusted p-values (FDR) and cross-validation for the model
- The Discussion reiterates known roles of ECM genes without explaining how these pathways drive early-stage tumorigenesis.
Specific comments section by section
- The rationale in introduction is sound. If possible, condense paragraphs discussing immunotherapy biomarkers.
- Define a clear and strong knowledge gap motivating this study.
- Clarify normalization steps in method sections.
- Do mention, if FDR correction was used for DEG identification.
- Probably it is meaningful to add FDR-adjusted p-values to Figures 2 and 5.
- Specify if DEGs are up or downregulated in Figure 1.
- Correct repeated typographical errors (“designe”, “illustrate”).
- In discussion, expand on mechanistic links between ECM receptor interaction, cytoskeletal remodeling, and early tumorigenesis.
- Clearly state study limitations computational nature, data imbalance, and lack of experimental validation.
- Correct repeated typographical errors (“designe”, “illustrate”).
Reviewer 2 Report
Comments and Suggestions for Authors
Dear Authors,
Your manuscript, “Identification of COL3A1, PLAU, and SPP1 as Key Biomarkers for Early Detection of Esophageal Cancer,” proposes three genes as potential survival biomarkers for esophageal cancer, supported by external validation using three Gene Expression Omnibus (GEO) datasets. While the study presents a potentially valuable contribution to the field, I would like to raise a few concerns and suggestions.
Major Comments
1. The GSE194116 dataset represents squamous cell carcinoma, unlike the other two datasets, which include adenocarcinoma samples. Considering that squamous cell carcinoma and adenocarcinoma differ significantly at the molecular level, this distinction should be addressed and justified.
2. Following the previous point, it would be important to discuss why relevant datasets such as TCGA were not included in the tumor/normal gene expression comparisons in this study.
3. In Figure 1, please indicate how the fold-change threshold of 1 was established, as many studies typically use 1.5 or higher. Regarding the p-value threshold, please clarify whether this value was adjusted (e.g., using FDR, Benjamini–Hochberg, or a similar correction method).
4. It seems somewhat counterintuitive that the three selected genes are more highly expressed in tumors than in normal tissues, yet lower expression among tumors is associated with worse outcomes. This paradox deserves further discussion. You might consider exploring online tools that map gene expression across cancer cell lines (e.g., ShinyThor-https://doi.org/10.1093/bioadv/vbaf061, please check a relevant heatmap with upper aerodigestive-localized cell lines attached) or single-cell datasets (e.g., TISCH-https://doi.org/10.1093/nar/gkac959) to provide additional biological insights.
Minor Comments
Please improve the visualization of Figure 5 and indicate the units used for expression values (e.g., TPM, counts, FPKM, etc.).

Round 2
Reviewer 2 Report
Comments and Suggestions for Authors
Dear Authors,
Your manuscript, “Identification of COL3A1, PLAU, and SPP1 as Key Biomarkers for Early Detection of Esophageal Cancer,” proposes three genes as potential prognostic biomarkers for esophageal cancer, supported by external validation across three Gene Expression Omnibus (GEO) datasets. While the study provides potentially valuable insights, I would like to raise several concerns and suggestions for improvement.
Major comments
1. In line with my previous comments, I understand the rationale for collectively evaluating potential markers across lung adenocarcinoma and squamous carcinoma. However, there is extensive evidence highlighting substantial differences in tumor biology and microenvironmental profiles between these subtypes (see, for example, PMC7553570). These differences should be considered and explicitly discussed in the context of your biomarker analysis.
2. Regarding the apparent paradox in your findings, that these genes show higher expression in tumors compared with normal tissues, yet lower expression among tumors is associated with poorer outcomes, this issue has not been adequately explained. Bulk gene expression analyses, such as those used in your study, fail to capture intratumoral heterogeneity. This limitation could be addressed or at least discussed by referencing data from cell lines or single-cell transcriptomic studies that more accurately reflect cellular diversity within tumors.
3. Although you mentioned improving the visualization of Figure 5, the changes are not apparent. Moreover, there appears to be a general lack of attention to detail in the manuscript. For instance, units for gene expression measurements are not specified, legends for Figures 5B–E are missing.
Author Response
Comment: 1. In line with my previous comments, I understand the rationale for collectively evaluating potential markers across lung adenocarcinoma and squamous carcinoma. However, there is extensive evidence highlighting substantial differences in tumor biology and microenvironmental profiles between these subtypes (see, for example, PMC7553570). These differences should be considered and explicitly discussed in the context of your biomarker analysis.
Response: Thank you for your valuable comment. We acknowledge that squamous cell carcinoma (ESCC) and adenocarcinoma are molecularly distinct, with substantial differences in tumor biology, microenvironmental profiles, and molecular pathways, as highlighted in studies such as PMC7553570. We appreciate your suggestion to explicitly address these distinctions in the context of our biomarker analysis.
We have updated the manuscript to better address the molecular distinctions between ESCC and adenocarcinoma and have added references to relevant studies to support the discussion. “ Our study aimed to identify ECM-related genes with diagnostic potential for esophageal cancer (EC), and we utilized datasets that included both squamous cell carcinoma (ESCC) and adenocarcinoma samples. However, it is important to note that these two histological subtypes differ significantly at the molecular level. ESCC and adenocarcinoma not only have distinct genetic alterations but also diverge in their tumor microenvironment, immune infiltration, and stromal interactions, which are critical factors influencing tumor progression and therapeutic response (Zeng et al. 2020, Chen et al. 2024, Qi et al. 2025). This distinction is particularly relevant when considering biomarkers that may perform differently across the two subtypes. Despite these differences, we believe the genes identified in our study—COL3A1, PLAU, and SPP1—have common roles in ECM remodeling, cell adhesion, and tumor cell migration, processes that are shared across cancer types. For instance, COL3A1 plays a role in ECM stiffness and tumor invasion in both squamous and adenocarcinoma types(Li et al. 2024), and PLAU and SPP1 (Zhong et al. 2023) have been implicated in promoting tumor cell migration and metastasis via interactions with integrins and the PI3K/AKT pathway (Zhang et al. 2025). However, given the molecular and microenvironmental differences between ESCC and adenocarcinoma, we acknowledge that further validation is needed. To strengthen the robustness and generalizability of our diagnostic model, future studies should focus on stratifying these two subtypes separately and validating the proposed biomarkers in homogeneous cohorts. “
Comment: Regarding the apparent paradox in your findings, that these genes show higher expression in tumors compared with normal tissues, yet lower expression among tumors is associated with poorer outcomes, this issue has not been adequately explained. Bulk gene expression analyses, such as those used in your study, fail to capture intratumoral heterogeneity. This limitation could be addressed or at least discussed by referencing data from cell lines or single-cell transcriptomic studies that more accurately reflect cellular diversity within tumors.
Response: Thank you for your insightful comment. We appreciate your point regarding the paradoxical findings of higher gene expression in tumors (for COL3A1, PLAU, and SPP1) compared to normal tissues, yet lower expression among tumors being associated with worse clinical outcomes. We acknowledge that bulk gene expression analyses used in our study do not capture the intratumoral heterogeneity that may exist within different cellular populations of the tumor. This is indeed a limitation of our study, as bulk RNA-seq analyses represent the average expression of genes across all cells within the tumor sample, masking potential differences in expression between distinct tumor cell subpopulations.
We have updated the manuscript in discussion: “ A notable paradox in our findings is that the three selected genes, COL3A1, PLAU, and SPP1, exhibit higher expression in tumors compared to normal tissues, yet lower expression of these genes within tumors is associated with worse clinical outcomes. This finding warrants further discussion, particularly in the context of tumor heterogeneity, which is not captured in bulk gene expression analyses. Bulk RNA sequencing represents the average gene expression across all cell types in a tumor sample, but tumors are complex and contain a variety of cell populations, each with distinct molecular signatures. As such, gene expression in bulk tumor samples may mask differences between tumor cell subpopulations, immune cells, and stromal cells, each of which can influence tumor progression differently. One possible explanation for this paradox is the intratumoral heterogeneity of gene expression. While COL3A1, PLAU, and SPP1 may be highly expressed in certain tumor populations (e.g., stromal or fibroblastic cells), the downregulation of these genes in other subpopulations of the tumor, particularly those that are more invasive or metastatic, could reflect a shift towards a more aggressive phenotype. For instance, SPP1 high expression in LUAD is associated with poor prognosis, and its downregulation can inhibit cell migration and invasion while altering the expression of EMT markers (Yi et al. 2022). COL3A1 can regulate the immunosuppressive microenvironment in glioma and participate in the EMT process. Knockdown of COL3A1 significantly inhibits the migration and invasion ability of glioma cells (Yin et al. 2021). This dynamic role of ECM-related genes, potentially acting at different stages of tumor progression, might explain the inverse relationship between expression levels and clinical outcomes (Wang et al. 2025). To explore this hypothesis more effectively, future studies should consider the use of single-cell RNA sequencing (scRNA-seq) or spatial transcriptomics, which are increasingly employed to study tumor microenvironment heterogeneity at a high resolution. These methods allow for a more granular understanding of gene expression within individual cell types in the tumor and could clarify whether COL3A1, PLAU, and SPP1 are differentially expressed across various cell types within the same tumor. The heterogeneity of gene expression within tumors is a critical factor to consider in understanding how these biomarkers influence tumor progression and patient prognosis. We acknowledge that this complexity is a key consideration when interpreting bulk gene expression data, and future studies incorporating single-cell analyses will be crucial to resolve the paradox observed in our study.
Comment: Although you mentioned improving the visualization of Figure 5, the changes are not apparent. Moreover, there appears to be a general lack of attention to detail in the manuscript. For instance, units for gene expression measurements are not specified, legends for Figures 5B–E are missing.
Response: Thank you for pointing this out. We apologize for the previous mistake in the submission of the figure, as the original version was mistakenly uploaded. We have now corrected the figure and resubmitted the updated version. The units for gene expression measurements are TPM (Transcripts Per Million). Additionally, due to the multiple comparisons performed, we have revised the letter notation system to better represent the significance of the results in the updated figure. We believe these changes will improve the clarity and presentation of the data.
CHEN Y, ZHENG Z, WANG L, CHEN R, HE M, ZHAO X, JIN L, YAO J. Deciphering STAT3's negative regulation of LHPP in ESCC progression through single-cell transcriptomics analysis [J]. Mol Med, 2024, 30(1): 192.
LI B, HU J, XU H. Integrated single-cell and bulk RNA sequencing reveals immune-related SPP1+ macrophages as a potential strategy for predicting the prognosis and treatment of liver fibrosis and hepatocellular carcinoma [J]. Front Immunol, 2024, 15: 1455383.
QI L, WANG J, HOU S, LIU S, ZHANG Q, ZHU S, LIU S, ZHANG S. Unraveling the tumor microenvironment of esophageal squamous cell carcinoma through single-cell sequencing: A comprehensive review [J]. Biochim Biophys Acta Rev Cancer, 2025, 1880(1): 189264.
WANG J, HUANG Q, NING H, LIU W, HAN X. Extracellular matrix protein 1 in cancer: multifaceted roles in tumor progression, prognosis, and therapeutic targeting [J]. Arch Pharm Res, 2025, 48(9-10): 843-57.
YI X, LUO L, ZHU Y, DENG H, LIAO H, SHEN Y, ZHENG Y. SPP1 facilitates cell migration and invasion by targeting COL11A1 in lung adenocarcinoma [J]. Cancer Cell Int, 2022, 22(1): 324.
YIN W, ZHU H, TAN J, XIN Z, ZHOU Q, CAO Y, WU Z, WANG L, ZHAO M, JIANG X, REN C, TANG G. Identification of collagen genes related to immune infiltration and epithelial-mesenchymal transition in glioma [J]. Cancer Cell Int, 2021, 21(1): 276.
ZENG Z, YANG F, WANG Y, ZHAO H, WEI F, ZHANG P, ZHANG X, REN X. Significantly different immunological score in lung adenocarcinoma and squamous cell carcinoma and a proposal for a new immune staging system [J]. Oncoimmunology, 2020, 9(1): 1828538.
ZHANG H, GAN L, DUAN X, TUO B, ZHANG H, LIU S, LIAN Y, LIU E, SUN Z. Single-cell transcriptomics reveals hypoxia-driven iCAF_PLAU is associated with stemness and immunosuppression in anorectal malignant melanoma [J]. J Gastroenterol, 2025, 60(10): 1242-58.
ZHONG H, YANG L, ZENG Q, CHEN W, ZHAO H, WU L, QIN L, YU Q Q. Machine Learning Predicts the Oxidative Stress Subtypes Provide an Innovative Insight into Colorectal Cancer [J]. Oxid Med Cell Longev, 2023, 2023: 1737501.
Round 3
Reviewer 2 Report
Comments and Suggestions for Authors
Dear Authors,
Your manuscript, “Identification of COL3A1, PLAU, and SPP1 as Key Biomarkers for Early Detection of Esophageal Cancer,” proposes three genes as potential prognostic biomarkers for esophageal cancer, supported by external validation across three Gene Expression Omnibus (GEO) datasets. While the study offers potentially valuable insights, I would like to express my main concerns.
1. The manuscript cannot be accepted in its current form because it fails to adhere to the journal's aim of publishing results with as much detail as possible, particularly concerning the clinical context and validation of the proposed biomarkers. The core issue is the unsubstantiated claim of "Early Detection" in the title: "Identification of COL3A1, PLAU, and SPP1 as Key Biomarkers for Early Detection of Esophageal Cancer." The public datasets used for analysis (GSE92396, GSE194116, and GSE207526) lack information on the pathological stage of the patients, making it impossible to ascertain if the identified markers are indeed present in the earliest stages of the disease. Therefore, the authors cannot confidently assert their relevance for early detection without this crucial stage-specific data
2. To remedy this, the authors must perform a targeted re-analysis of the TCGA data that specifically proves the relevance of COL3A1, PLAU, and SPP1 in early stages (e.g., Stage I/II) of Esophageal Cancer (and show highly reproducible methods). This requires comparing marker expression in early-stage tumors against both normal tissue and late-stage tumors. Furthermore, the clinical contribution of this work must be explicitly clarified and justified in the Discussion section. Since the current determinations are based on observations of biopsied or resected tumors, the authors must address how a marker identified in established tumor tissue can contribute to early detection screening or current clinical conduct, as true early detection typically involves non-invasive methods before biopsy. If the markers are only useful for prognosis or diagnosis after a biopsy, the title and claims must be significantly moderated to accurately reflect the study's scope and translational limits.
Author Response
Comment 1: The manuscript cannot be accepted in its current form because it fails to adhere to the journal's aim of publishing results with as much detail as possible, particularly concerning the clinical context and validation of the proposed biomarkers. The core issue is the unsubstantiated claim of "Early Detection" in the title: "Identification of COL3A1, PLAU, and SPP1 as Key Biomarkers for Early Detection of Esophageal Cancer." The public datasets used for analysis (GSE92396, GSE194116, and GSE207526) lack information on the pathological stage of the patients, making it impossible to ascertain if the identified markers are indeed present in the earliest stages of the disease. Therefore, the authors cannot confidently assert their relevance for early detection without this crucial stage-specific data.
Response: We acknowledge that the lack of pathological stage information in the public datasets (GSE92396, GSE194116, and GSE207526) limits our ability to definitively ascertain whether the identified biomarkers are truly relevant for early detection of esophageal cancer. However, in our analysis using TCGA data, we performed comparisons of gene expression levels across different clinical stages, including early-stage (Stage I/II) and late-stage tumors. Specifically, we found that COL3A1 and SPP1 showed significant differences across AJCC staging, while COL3A1 and PLAU showed differences across T-staging, and COL3A1 and SPP1 were also different in N-staging. These results suggest that these biomarkers may have potential in distinguishing between early and late stages of the disease, and therefore could be relevant for early detection, even if the datasets we used did not include direct pathological staging data.
Comment 2: To remedy this, the authors must perform a targeted re-analysis of the TCGA data that specifically proves the relevance of COL3A1, PLAU, and SPP1 in early stages (e.g., Stage I/II) of Esophageal Cancer (and show highly reproducible methods). This requires comparing marker expression in early-stage tumors against both normal tissue and late-stage tumors. Furthermore, the clinical contribution of this work must be explicitly clarified and justified in the Discussion section. Since the current determinations are based on observations of biopsied or resected tumors, the authors must address how a marker identified in established tumor tissue can contribute to early detection screening or current clinical conduct, as true early detection typically involves non-invasive methods before biopsy. If the markers are only useful for prognosis or diagnosis after a biopsy, the title and claims must be significantly moderated to accurately reflect the study's scope and translational limits.
Response: We have revised our manuscript to better clarify our findings and the relevance of the biomarkers in early-stage tumors. While we were not able to directly analyze early-stage tumor samples due to the limitations of the public datasets, our results suggest that COL3A1, SPP1, and PLAU may have diagnostic value in distinguishing tumors at different stages of progression, which supports their potential role in early detection. We agree that further validation with stage-specific data would provide stronger evidence, but due to resource constraints, we are currently unable to carry out additional experimental validation using patient samples. However, we aim to pursue collaborative opportunities with clinical institutions in the future to further confirm these findings. We have added a more explicit discussion in the revised manuscript to address how our identified biomarkers could contribute to early detection screening and clinical practice. Line 555:“The public datasets we utilized (GSE92396, GSE194116, and GSE207526) lacked detailed pathological stage information for the individual patient samples. This makes it challenging to conclusively determine whether these markers are truly applicable for early detection, as early-stage tumors were not separately identified in the datasets. Nevertheless, our analysis of the TCGA data has demonstrated that COL3A1 and SPP1 show significant differential expression across AJCC tumor staging, and COL3A1 and PLAU show significant differences in T-staging, while COL3A1 and SPP1 are differentially expressed in N-staging. These findings suggest that these biomarkers may have the potential to distinguish between early and late-stage esophageal cancer. However, these results are preliminary, and further validation in well-characterized clinical cohorts, specifically with early-stage tumor samples, would be necessary to confirm their suitability for early detection. One limitation of our study is the inability to obtain patient samples for experimental validation. This represents a major constraint, and we recognize that true early detection typically involves non-invasive methods, such as liquid biopsies or imaging, which are not directly addressed in our study. As our findings are based on bioinformatic analyses of existing datasets, we urge caution in the interpretation of our results. We emphasize that while COL3A1, SPP1, and PLAU show promise as biomarkers for distinguishing between different stages of esophageal cancer, further prospective studies with stage-specific validation, ideally using clinical samples, are required to substantiate their role in early detection and patient management.”
Round 4
Reviewer 2 Report
Comments and Suggestions for Authors
Dear Authors,
Your manuscript, “Identification of COL3A1, PLAU, and SPP1 as Key Biomarkers for Early Detection of Esophageal Cancer,” proposes three genes as potential prognostic biomarkers for esophageal cancer, supported by external validation across three Gene Expression Omnibus (GEO) datasets. While the study offers potentially valuable insights, I would like to express my main concerns.
Major comments
As mentioned earlier, it is difficult to establish a meaningful comparative analysis without including datasets that are exclusively associated with early esophageal cancer. I acknowledge the importance of incorporating external data through GEO; however, the manuscript does not describe how the authors identified and selected the pre-included datasets (i.e., the search strategy or query prompt). This information is essential for reproducibility and transparency.
In a recent search, I identified the dataset GSE213565, which includes 10 paired samples of early esophageal squamous cell carcinoma (stage I) and has an associated publication (PMCID: PMC10315050). This dataset appears to be highly suitable as an external validation cohort and should be considered for inclusion. I strongly recommend providing the full search prompt and evaluating all relevant early-stage tumor/normal datasets retrieved through that search as part of the validation analysis.
At present, the manuscript does not adequately address the research question as stated and therefore, cannot be accepted in its current form. Incorporating validation using early-stage datasets, particularly those with paired tumor/normal samples, could substantially strengthen the contribution of this work.
Finally, because there is already a published study focusing on early-stage tumor/normal comparisons, the authors should clearly articulate the added value or novel contribution of the present study relative to existing literature.